# Diagnostic Efficacy of Rectal Suction Biopsy with Regard to Weight in Children Investigated for Hirschsprung’s Disease

**DOI:** 10.3390/children9020124

**Published:** 2022-01-18

**Authors:** Emma Fransson, Christina Granéli, Kristine Hagelsteen, Louise Tofft, Mette Hambraeus, Rodrigo Urdar Munoz Mitev, David Gisselsson, Pernilla Stenström

**Affiliations:** 1Department of Pediatric Surgery, Institution of Clinical Sciences, Skåne University Hospital, Lund University, 22185 Lund, Sweden; emma.fransson.3843@med.lu.se (E.F.); christina.graneli@med.lu.se (C.G.); kristine.hagelsteen@med.lu.se (K.H.); louise.tofft@med.lu.se (L.T.); mette.hambraeus@skane.se (M.H.); 2Clinical Genetics and Pathology, Laboratory Medicine Skåne, University Hospital, 22185 Lund, Sweden; Rodrigo.MunozMitev@skane.se (R.U.M.M.); david.gisselsson_nord@med.lu.se (D.G.)

**Keywords:** children, Hirschsprung’s disease, rectal suction biopsy

## Abstract

Background/aim: Diagnostic efficacy, defined as the percentage of rectal suction biopsy (RSB) specimens sufficient enough to determine the absence of ganglia cells in Hirschsprung’s disease (HD) diagnosis, has been reported to be low, requiring repeated biopsies. The aim was to explore whether RSB diagnostic efficacy was influenced by the child’s weight and to ascertain whether RSB efficacy differed between aganglionic and ganglionic tissue. Materials and Methods: Efficacy analyses were conducted in a national HD-center’s register on children 0–15 kg, examined between 2011–2019. First-time RSB diagnostic efficacy was correlated to the children’s weight and final HD diagnosis. Results: Among the 84 children who had first-time RSB, the overall diagnostic efficacy was 85% (71/84). The efficacy was higher among children weighing less than the identified cut-off of 9.0 kg (89% in 0–9.0 kg versus 62% in 9.01–15.0 kg, *p* = 0.026). Among children diagnosed with HD, 96% (26/27) weighed 0–9.0 kg. In this weight group, the diagnostic efficacy was lower in aganglionosis compared to ganglionosis (77%; 20/26 versus 96%; 43/45), *p* = 0.045). Conclusions: The RSB diagnostic efficacy was significantly higher in children weighing less than 9.0 kg and was less in aganglionic compared to ganglionic tissue. Therefore, weight can be useful to predict RSB diagnostic efficacy.

## 1. Introduction

Hirschsprung disease’s (HD) is a developmental disorder of the enteric nervous system, characterized by the absence of ganglion cells in the myenteric and submucosal plexus in the bowel wall. For definitive HD diagnosis, a rectal biopsy is required [1]. In general, two different methods of collecting rectal biopsies are used: rectal suction biopsy (RSB) and full-thickness biopsy (FTB) [2]. Since the technique of RSB was first published in 1965, it has become the preferred diagnostic tool, mainly because it can be performed without anesthesia and suturing [3,4]. The main reported disadvantage of RSB has been its low efficacy due to superficial specimens without enough submucosa to confirm the absence of submucosal ganglion cells. In a systemic review, the overall diagnostic efficacy of RSB specimens was reported to be 90%, but in several studies, RSB efficacy has been reported to be even lower, causing diagnostic delays, morbidity, longer hospital stays, and higher costs of treatment [5,6,7,8,9,10]. It has been suggested that older children, due to their thicker submucosa, might be subjected to a higher risk of repeated RSB [11], but it is still controversial as to what age RSB should be replaced by FTB in order to reduce the risk of repeated examinations. Even though the mucosa’s and submucosa’s thicknesses might theoretically correlate closer to the child’s weight than age, several studies have only focused on efficacy with regard to the child’s age and not on weight [6,7,8,9,12,13,14,15]. Another potential factor that might influence the RSB diagnostic efficacy is the presence or absence of ganglion cells. This is because the bowel wall’s rigidity, i.e., its stiffness, might bring difficulties in the RSB reaching a deep enough level in the submucosa, as the rigidity might be increased if ganglion cells are absent in the submucosal plexa. However, this is just a theory and has never been studied in detail.

The study’s main aim was to explore whether the child’s weight influenced the RSB diagnostic efficacy, defined as sufficient specimens to determine the presence or absence of ganglion cells. The secondary aim was to evaluate whether RSB diagnostic efficacy differed between aganglionic and ganglionic tissue.

## 2. Materials and Methods

### 2.1. Patients and Data

This retrospective study was conducted at a national referral center for HD. Patients 0–15.0 kg undergoing diagnosis with RSB between July 2011 and June 2019 for suspected HD were identified from a local HD and biopsy register. The data were collected from medical charts and included symptoms, age and weight at time of biopsy, gestational week, number of biopsies, biopsy results, need for additional biopsies, complications, and the name of the pediatric surgeon who collected the biopsy. Complications of the biopsy procedures were registered according to the Clavien–Dindo classification [16]. The follow-up period was determined as the time from biopsy until the end of the study. Diagnostic efficacy was defined as the percentage of sufficient rectal biopsy specimens to determine the presence or absence of ganglion cells. The RSB diagnostic efficacy and complications were, for study purposes, analyzed only the first time that RSB was performed. The final diagnosis of HD was made when the surgical resection of an aganglionic specimen was performed.

### 2.2. Investigation Routines

Children admitted to the department of pediatric surgery for HD evaluation were investigated routinely according to the department’s HD diagnostic guidelines. These included registering HD characteristic symptoms, radiologic examination with colorectal enema, and rectal biopsy. In our study, HD-specific symptoms required for further investigation for HD were: (1) failure to pass meconium during the first 48 h of life; (2) the need for daily rectal washouts and/or enemas due to either outlet obstruction, including chronic difficulties in passing gases and feces despite soft fecal consistence, or constipation, including defecation fewer than two times per week. Early-onset symptoms were defined as symptoms presenting within 1 month of age. Symptoms were reported by proxy of the guardian.

The HD-specific pathological findings on colorectal contrast enema were a recto-sigmoidal quote of <1, dilated colon, and/or absence of a rectoanal inhibitory reflex [17].

### 2.3. Biopsy Technique and Staining

RSB was introduced at the department as the first-line diagnostic method in 2011 [18]. Over the study period 2011–2019, 13 pediatric surgeons performed the procedure.

RSB was performed at the outpatient clinic using the Rbi2^®^ suction instrument (Aus Systems, Charles Sturt, Australia). In line with local guidelines, all surgeons received education regarding the Rbi2-biopsy technique, and a set of three suction biopsies were then taken in the posterior rectum wall at a level of 1, 2, and 3 cm above the dentate line. Hemostasis was maintained using Spongostan^®^. Rectal catheterization was allowed 24 h after RSB. The children were observed in hospital up to 4 h after the procedure.

The biopsy specimens were fixed in formalin, embedded in paraffin, and stained with hematoxylin eosin to visualize the rectal tissue and ganglion cells, and subsequently stained immunohistochemically with calretinin and S100 to identify the nerve cells. Using standard, automated immunohistochemical methods, S100 protein was detected using a ready-to-use polyclonal antibody (Ventana, catalog no. 760-2523) and calretinin using the ready-to-use monoclonal antibody SP65 (Ventana, catalog no. 790-4467). Three pediatric pathologists performed all of the pathological analyses and reports. For study purposes, the biopsies were categorized into three groups for analyses: absence of ganglion cells, presence of ganglion cells, or inconclusive. The latter was when the pathologist could not determine the presence of ganglion cells due to low tissue quality or too superficial specimens in all the biopsies in the set.

### 2.4. Statistical Analysis

The analysis of the data was performed using SPSS version 27 (IBM Corp., Armonk, NY, USA). Non-parametric tests were used due to the limited number of patients. A Mann–Whitney U-test was applied for the analyses of the continuous data, and Fisher’s exact test and Pearson’s chi^2^ test for dichotomous or multiple comparisons. Spearman’s correlation test was used for age and weight matching. The RSB efficacy within the lower weight group was compared with the higher weight group. A cut-off weight for significantly higher RSB efficacy was set where a significance was identified comparing diagnostic efficacy per weight group. A *p*-value of less than 0.05 was considered to be significant.

## 3. Ethical Considerations

This study was ethically approved by the regional ethical review board (registration number 2017/191).

## 4. Results

### 4.1. Patients

A total of 84 children 0–15.0 kg who underwent first-time RSB in the diagnosis of HD were identified in the register (Table 1). All 84 children presented with HD-typical symptoms, and 80% (*n* = 67) presented with symptoms before 1 month of age. Overall, 90% (76/84) of the children selected for RSB had undergone a colorectal contrast enema prior to biopsy. All of the 27 children who were finally diagnosed with HD had abnormal contrast enemas. One child had anorectal manometry performed at another hospital before the RSB was performed at our department. The distribution of the cohort’s early onset symptoms, contrast enema outcomes, and final HD diagnosis are displayed in Figure 1.

### 4.2. Rectal Suction Biopsy (RSB) Data and Efficacy

The median number of RSB specimens per patient sent to the pathologists was three (range one to five). The diagnostic efficacy, defined as sufficient specimens to determine the presence or absence of ganglion cells, of first-time RSB was 85% (71/84) (Table 1). Children who did not have sufficient specimens in their biopsies for determining the absence or presence of ganglion cells (n = 13) underwent repeated biopsies, either RSB or FTB, until aganglionosis could be confirmed or revealed (Figure 2). None of the children who did not have rectal biopsy were shown to have HD within the study’s follow-up time of a median of 5 (1–9) years. The overall complication rate within 4 weeks was 2/84 (2%). The two children with complications after RSB weighed 4–5 kg and were readmitted within 24 h for bleeding, requiring blood transfusion and bleeding control under general anesthesia, corresponding to Clavien–Dindo 3b. None of them had aganglionosis. The diagnostic efficacy, i.e., obtaining accurate RSBs containing submucosa, ranged from 50–100% between the physicians (n = 13) who had performed the biopsies during the study period (*p* = 0.518).

A cut-off weight was defined at 9.0 kg, where a significant difference in diagnostic efficacy was identified between the lower and higher weight groups (Figure 3). The diagnostic efficacy was thus highest in children weighing below 9 kg compared to that in heavier children. No children weighing 12–15 kg (n = 4) had conclusive biopsies (Figure 3). Age and weight correlated positively (rho 0.763).

Aganglionosis was diagnosed in 32% (27/84) of the children. Of the children who either at first-time RSB (n = 20) or at repeated biopsies (n = 7) were found to have aganglionosis, 93% (25/27) weighed between 2–4 kg and 96% (26/27) weighed below the cut-off weight of 9 kg. In Figure 4, an overview of the distribution of aganglionosis, efficacy, and weights in all first-time RSB is presented.

The RSB diagnostic efficacy in first-time RSB was overall 74% (20/27) in children with aganglionosis and 89% (51/57) in children with a ganglionic specimen (*p* = 0.104). In the group of children weighing less than 9.0 kg (the cut-off weight), the diagnostic efficacy was 77% (20/26) for aganglionosis compared to 96% (43/45) with ganglionosis (*p* = 0.045). All patients with aganglionosis had this confirmed in the analyses of the resected specimens after HD surgery. None of the 51 children with first-time RSB analyses negative for aganglionosis were false-negative, i.e., no child presented later with HD during the follow-up period of a median of 5 (1–9) years.

## 5. Discussion

The main results of this study showed that the diagnostic efficacy of first-time RSB-specimen in children weighing up to 15 kg was associated with weight. A diagnostic efficacy cut-off could be identified at 9.0 kg, where a higher diagnostic efficacy was identified in children weighing less than 9 kg compared to those weighing over 9 kg. The diagnostic efficacy tended to be lower in the aganglionic specimen overall, and was significantly lower in aganglionic specimens in the group weighing less than 9 kg.

The results indicate that the chance of obtaining a successful analysis of first-time RSB is higher in children weighing less than 9.0 kg, and that it could be influenced by the ganglionic status of the biopsy specimen. This could have implications on information given to the care givers, bringing realistic expectations on the diagnostic efficacy within the RSB procedure. It might also serve as guidance for pediatric surgeons concerning the choice of biopsy method, where one could consider FTB in children with disease-specific HD symptoms and higher weights, especially if there is a need for repeated biopsy.

To the authors’ best knowledge, an association between weight and RSB efficacy has not been studied previously. Our results showed that age and weight correlated fairly positively, but since weight is variable between children of the same ages, and the bowel wall’s thickness and histoanatomic layers theoretically might differ with the child’s weight, although never proven, the latter could be a more precise measure compared to age in predicting RSB’s diagnostic efficacy. Our result, that RSB efficacy was lower in higher weight groups, is in line with other studies, in which some suggest replacing RSB with FTB in higher age groups such as in children aged 6, 12, and 36 months [7,12,19].

In addition, the diagnostic efficacy in the current study seemed to be influenced by aganglionosis, tending to be lower in aganglionic than in normo-ganglionic tissue. However, this finding was not significant within the present study’s cohort overall, but only in the group of children weighing below 9.0 kg. The decrease in RSB efficacy in children with higher weights might, speculatively, be attributed to a growth-associated increase in thickening and possibly rigidity of the intestinal mucosa. Increased rigidity has been suggested to cause a decrease in RSB efficacy, as in patients with chronic constipation and megarectums [7,11]. Our results, i.e., that RSB efficacy was lower in patients with an aganglionic bowel wall (74%), could support this theory, since a higher tissue rigidity could be expected in aganglionic tissue. In line with this theory, two studies reported that if an RSB was classified as inconclusive, the predictive value for eventually confirming HD was 79% and 83% in the respective studies [15,19]. If this reasoning is true, i.e., that the efficacy tends to be lower in aganglionosis, the aganglionic frequency in the study cohorts would influence the diagnostic efficacy outcome. In addition, a stricter selection procedure for biopsy might lower the diagnostic efficacy. In our study, the overall frequency of HD after finalized investigations was 32%, which is in the higher range of previously reported HD frequencies (10–43%) [5,8,9,10,12,13,14,15].

With regard to the diagnostic efficacy in HD investigations, it is important to consider the challenge to prove the absence of tissue, i.e., ganglion cells, instead of the presence, imposing an even greater demand on high-quality tissue and precise specimen handling when the suspicion of aganglionosis is higher. Previous studies have stated that specimen handling, staining procedures, and the expertise of the pathologist might influence the diagnostic efficacy of RSB [15,19]. However, this has never, to our best knowledge, been studied specifically. When our pediatric pathologists were able to identify the submucosa in the biopsy samples, all aganglionic biopsy specimens were confirmed by resected bowel specimens. During the follow-up period, no child with ganglionosis in biopsies was found to have HD. These high sensitivity and specificity results are in accordance with some previous studies [19,20], while two systemic reviews reported marginally lower RSB mean sensitivities (93% and 97%) and specificities (98% and 99%) [5,21]. The differences in outcome might be due to different definitions of efficacy, sensitivity/specificity, staining techniques, or follow-up periods.

In the study cohort, 96% of the children with a final diagnosis of HD presented with early symptoms, and all showed pathology on colorectal contrast enema, while none of the children with late-presenting symptoms and a normal recto-cologram were found to have HD. None of the children who did not have rectal biopsy were found to have HD within the study’s follow-up time of a median of 5 (1–9) years. This reflects that careful medical history and pre-defined definitions of pathology on radiologic examinations could safely be used to reduce the unnecessary use of rectal biopsy, thereby minimizing the children’s discomfort, complications, and pathological work up. However, for the purpose of reducing the frequency of RSB with high safety, further studies are needed, including prospective inclusion according to a predefined weight and radiologic algorithm.

A strength of this study is that the analyses included all children in the region who underwent investigation for HD during the entire study time, without any drop out. In addition, the same pathohistological staining (hematoxylin eosin, calretinin, and S100) was used throughout the study period. The main limitation was that most patients (80%) were neonates with a median weight of 4 kg, which could have skewed the data significantly. Another weakness is the small number of patients, which limited the statistical power. In addition, compliance with the guidelines regarding level of biopsing could only be confirmed in the later years when a strict documentation of biopsy levels was established. The scientific quality could improve, and results be more generalizable, by including more specimens and institutions. Since RSB is used in the vast majority of institutions worldwide, a multicenter study would be feasible and is warranted. This is one way forward to reveal whether the rectal biopsy efficacy could be improved by selecting children on the basis of their weight for either RSB or FTB.

## 6. Conclusions

This first study of RSB diagnostic efficacy with regard to weight showed that the RSB diagnostic efficacy was higher in children weighing 0–9 kg than in heavier children. The diagnostic efficacy tended to be lower in aganglionic tissue. This could have implications for the caregivers’ and professionals’ expectations of the diagnostic efficacy of first-time RSB, and provide a basis for further procedure developments. According to the results, one could consider RSB only in children with lower weights. To increase the RSB’s overall diagnostic efficacy, methods for improving tissue handling and increasing the tissue quality and submucosal content seem relevant to implement.

## Figures and Tables

**Figure 1 children-09-00124-f001:**
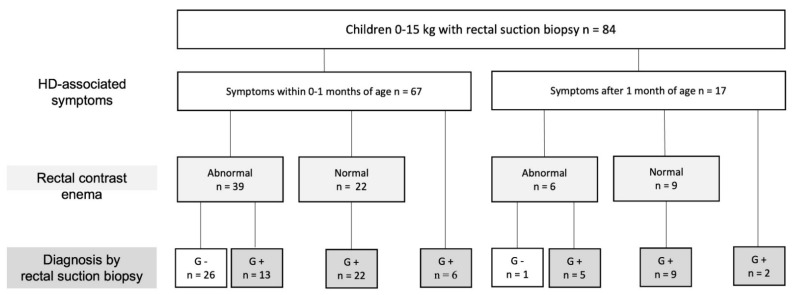
Diagnostic work up for children (n = 84) investigated for Hirschsprung’s disease (HD) with rectal suction biopsy. Aganglionosis (G−) and ganglionosis (G+).

**Figure 2 children-09-00124-f002:**
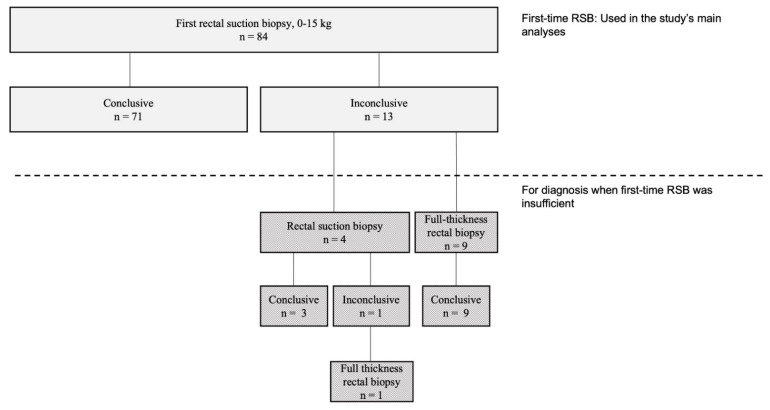
Flowchart of outcome of first-time rectal suction biopsy (RSB) in patients examined for Hirschsprung’s disease (HD).

**Figure 3 children-09-00124-f003:**
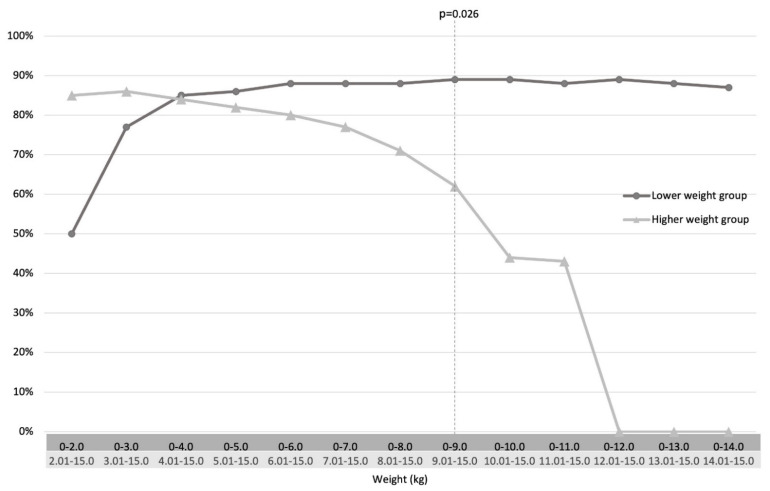
Rectal suction biopsy diagnostic efficacy, defined as sufficient specimen including submucosa for histopathological analyses, compared between weight groups. Cut-off was set when the group difference reached a significant value.

**Figure 4 children-09-00124-f004:**
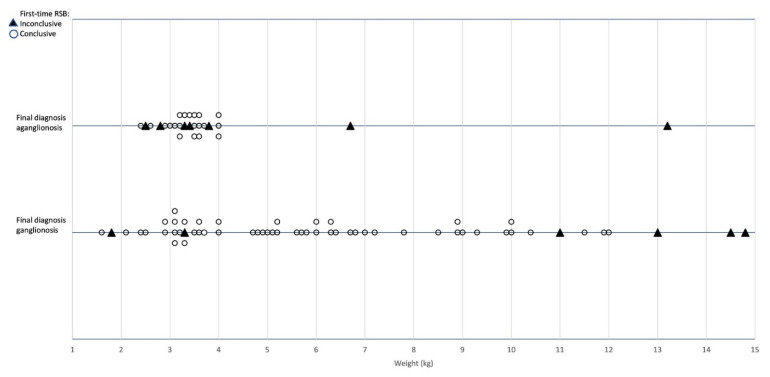
Descriptive overview of all 84 rectal suction biopsies with regard to weight, conclusive or inconclusive result on first biopsy, and final diagnosis of aganglionosis/ganglionosis.

**Table 1 children-09-00124-t001:** Comparison of characteristics of 84 children with suspected Hirschsprung’s disease (HD) examined with first-time rectal suction biopsy (RSB), with conclusive (specimen with identified submucosa) and inconclusive (specimen without identified submucosa) biopsies. n (%), median (min-max).

	All First Time RSB n = 84	First-Time RSB with Conclusive Diagnostics n = 71	First-Time RSB with Inconclusive Diagnostics n = 13	*p*-Value
Gender (female)	29 (35)	26 (37)	3 (23)	
Age at biopsy (days)	42 (1–1345)	58 (1–860)	26 (2–1345)	0.508 ^a^
Weight at biopsy (kg)	4.0 (1.6–14.8)	4.0 (1.6–12.0)	3.8 (1.8–14.8)	0.504 ^a^
Diagnostic efficacy at first biopsy	71 (85)			
Diagnosed with HD (n = 27)	27 (32)	20 (28)	7 (54)	0.104 ^b^

^a^ Mann–Whitney U-test, two-tailed. ^b^ Fisher exact test, two-sided.

## Data Availability

The complete data is available at the hospital’s diagnose based database. Please contact pernilla.stenstrom@med.lu.se for further information.

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
