# Peer review of "Diagnostic Efficacy of Rectal Suction Biopsy with Regard to Weight in Children Investigated for Hirschsprung’s Disease"

_children, 2022, doi:10.3390/children9020124_

Round 1
Reviewer 1 Report
This is a typical, single-center retrospective study, nicely written and easy to follow. The study is well conducted and implemented a formal statistical analysis. However, there is a problem with the hypothesis. Issues with the RSB have been described in numerous studies, as the authors state in their introduction. The problem is inherent in the technique. And if the method is not conducted correctly, there is a high risk of getting insufficient tissue for the pathologist. The main argument for the RSB is that it can be done bedside, whereas a proper rectal biopsy done in the operation theatre will require anesthesia with the known risk factors. However, the latter usually provides more tissue. Therefore, it is considered more efficient. Given the considerable impact on the baby's and parents' life once diagnosed with Hirschsprung Disease, one should be highly careful which method will be employed to reveal the correct diagnosis safely and efficiently way.
1) Can the authors provide an age: weight matching? It would be interesting to see.
2) Did the authors change their practice according to the study's findings?
3) The authors state in the conclusion section that their findings could have implications for the caregiver's and professionals' expectations of the diagnostic efficacy of first-time RSB. What do they mean exactly? Do they recommend an FTB as a primary approach beyond the weight of 9 kg?
Author Response
Dear editor and reviewer
Thanks a lot for your valuable comments and questions. We have answered in the table "Reply to reviewers" and changed accordingly in the re-submitted "Revised manuscript" where changes are marked.
Best regards
Pernilla Stenström

Reviewer 2 Report
In this study, Emma Fransson, et al. studied whether the child’s weight influenced the rectal suction biopsy (RSB) diagnostic efficacy for Hirschsprung disease. They conclude that RSB diagnostic efficacy was significantly higher in children weighing less than 9.0 kg and was less in aganglionic tissue compared to ganglionic tissue, and weight can be useful to predict RSB diagnostic efficacy. Authors findings are significant. However, majority of patients (80%) were neonates with a median weight of 4 kg, which could have skewed the data significantly and affected the validity of this study. I would also like to know if any of their patients had anorectal manometry prior to RSB and if the manometry findings correlated with RSB results.
Author Response

(The authors gave the same response as above.)

Round 2
Reviewer 1 Report
The article has been greatly improved; yet, I am disappointed that the scientific merit remains average, given that RSB is employed in the vast majority of institutions worldwide.
Finally, the authors' diligence and hard work should be recognised, and the research highlights how tough it is to provide considerable support in this relevant problem. The article, on the other hand, compiles and discusses RSB vs. FTB accordung to weight but fails to provide any further information other than that RSB diagnostic effectiveness was much higher in youngsters weighing less than 9.0 kg.
Author Response
Dear reviewer
We are grateful for the careful review, and have changed accordingly.
Best regards
Pernilla Stenström

Reviewer 2 Report
The authors have addressed all of my concerns with the original manuscript. The revised manuscript is ready for publication.
Author Response
Dear reviewer
We are grateful for your careful review of our manuscript.
Best regards
Pernilla Stenström
